# Combining Electrostimulation with Impedance Sensing to Promote and Track Osteogenesis within a Titanium Implant

**DOI:** 10.3390/biomedicines11030697

**Published:** 2023-02-24

**Authors:** Nadja Engel, Michael Dau, Vivien Engel, Denise Franz, Fabian Klemmstein, Christiane Thanisch, Jürgen F. Kolb, Marcus Frank, Armin Springer, Rüdiger Köhling, Rainer Bader, Bernhard Frerich, Nadine Wiesmann, Diana Heimes, Peer W. Kämmerer

**Affiliations:** 1Department of Oral and Maxillofacial Surgery, Facial Plastic Surgery, Rostock University Medical Center, Schillingallee 35, 18057 Rostock, Germany; 2Oscar Langendorff Institute of Physiology, Rostock University Medical Center, Gertrudenstrasse 9, 18057 Rostock, Germany; 3Institute of Physics, University of Rostock, Albert-Einstein-Str. 23-24, 18059 Rostock, Germany; 4ibidi GmbH, Lochhamer Schlag 11, 82166 Gräfelfing, Germany; 5ZIK Plasmatis, Leibniz Institute for Plasma Science and Technology (INP), a Member of the Leibniz Research Alliance Leibniz Health Technology, Felix-Hausdorff-Str. 2, 17489 Greifswald, Germany; 6Medical Biology and Electron Microscopy Centre, University Medical Center Rostock, Strempelstraße 14, 18057 Rostock, Germany; 7Department Life, Light and Matter, University of Rostock, Albert-Einstein-Str. 25, 18059 Rostock, Germany; 8Department of Orthopedics, Rostock University Medical Center, Doberaner Str. 142, 18057 Rostock, Germany; 9Department of Oral and Maxillofacial Surgery, Facial Plastic Surgery, University Medical Center Mainz, Augustusplatz 2, 55131 Mainz, Germany

**Keywords:** electrical stimulation, titanium implants, impedance, osteogenesis, ECIS

## Abstract

(1) Background: Electrical stimulation is a promising alternative to promote bone fracture healing but with the limitation of tracking the osteogenesis progress in vivo. To overcome this issue, we present an opportunity to combine the electrical stimulation of a commercial titanium implant, which promotes osteogenesis within the fracture, with a real-time readout of the osteogenic progress by impedance sensing. This makes it possible to adjust the electrical stimulation modalities to the individual patient’s fracture healing process. (2) Methods: In detail, osteogenic differentiation of several cell types was monitored under continuous or pulsatile electrical stimulation at 0.7 V AC/20 Hz for at least seven days on a titanium implant by electric cell-substrate impedance sensing (ECIS). For control, chemical induction of osteogenic differentiation was induced. (3) Results: The most significant challenge was to discriminate impedance changes caused by proliferation events from those initiated by osteogenic differentiation. This discrimination was achieved by remodeling the impedance parameter Alpha (α), which increases over time for pulsatile electrically stimulated stem cells. Boosted α-values were accompanied by an increased formation of actin stress fibers and a reduced expression of the focal adhesion kinase in the cell periphery; morphological alterations known to occur during osteogenesis. (4) Conclusions: This work provided the basis for developing an effective fracture therapy device, which can induce osteogenesis on the one hand, and would allow us to monitor the induction process on the other hand.

## 1. Introduction

For decades, the guidance of stem cells to repair or replace non-functional or destroyed tissue has been the main challenge in many fields of tissue engineering and regenerative medicine. As adipose tissue represents an exciting source of adult stem cells with the ability to differentiate along multiple lineage pathways, a focus went toward using adipose-derived mesenchymal stem cells (ASCs) [1,2]. Considering the increased incidence of obesity in industrialized countries, subcutaneous adipose tissue is abundant and readily accessible. For example, hundreds of thousands of liposuction surgeries were undertaken annually, yielding 100 mL to two liters of lipoaspirate tissue per intervention. However, stem cells are mainly insufficient to repair demanding (“critical size”) osseous defects. Here, a combination of a scaffold (especially titanium or biocompatible bone substitute materials) and stem cell enrichment is required for an ideal regenerative approach. Regenerative processes around the implant system can be monitored mainly using invasive procedures such as obtaining bone samples from the implant site for histological evaluation or (3D) radiographic methods [3,4]. Though the histological approach requires the destruction of the peri-implant environment and radiological imaging can be impaired by artifact formation in the case of metallic implants. A potential solution could be to develop an implant system that simultaneously stimulates the osteogenic differentiation of stem cells and offers a non-invasive read-out of the actual differentiation processes. For this purpose, we choose a combination of an electrically stimulating titanium implant to direct ACSs and bone-marrow-derived stem cells (BMSC) into the osteogenic differentiation in combination with electric cell-substrate impedance spectroscopy (ECIS) as pioneered by Giaever and Keese [3,4,5] to monitor the osteogenic processes [6,7]. 

Recently, our group evaluated electrical stimulation parameters to improve ASC proliferation in vitro [8]. Here, continuous stimulation under the setting of 0.7 V AC/20 Hz revealed a proliferation increase within the first three days. A longer duration of electrical stimulation forced metabolic arrest and apoptosis initiation. We applied pulsatile stimulation, which resulted in enhanced osteocalcin and collagen I expression using periods of 3 × 45 min per day with a 225 min break in between. These pulsatile stimulation settings have favored osteogenic differentiation and increased bone formation [9,10]. Therefore, in the present work, we compared continuous and pulsatile electrical stimulation on stem cells of adipose origin (ASC) and bone-marrow-derived stem cells (BMSC), and primary osteoblasts (POB). The osteogenic process was monitored using impedance sensing. One of the advantages of this sensing method is that impedance alterations of cells seeded on a gold electrode array can be tracked in real time by applying a very weak (<1 µA) alternating current (I_ECIS_). This technique has been used in several experimental approaches such as cell motility studies [4,11], drug response, analyzing cancer cell lines [12], and assessment of barrier function [13]. Furthermore, the specific approach of electric cell-substrate impedance sensing (ECIS) allows for determining morphology-related electrical parameters that can be derived from suitable models for the observed impedance responses. The ECIS approaches were mainly applied in studies dealing with endothelial cells to investigate barrier function-related parameters [14,15,16]. The cell coverage and barrier resistance values are summarized in the parameter R_b_. However, in endothelial and epithelial cells, the membrane creates capacitive behavior by storing charges at either side of their hydrophobic core. Therefore, in addition to the resistance of barrier-forming cell–cell junctions (R_b_), alterations in the cell layer membrane capacitance (C_m_) can be analyzed. C_m_ can be used as an indicator for the cell layer integrity, which occurs, for example, during differentiation processes [6,17]. It is important to note that high frequencies (>60 kHz) are required for describing the integrity and dielectric properties of the cell membrane [18,19,20]. The current I_ECIS_ can capacitively couple through the plasma membrane at high frequencies (>32,000 Hz). Lower frequencies (≤4000 Hz) are used to model cell-substrate adhesion and cell–cell tight junctions because the current I_ECIS_ is forced to move under or between the cells. Therefore, it is essential to conduct multifrequency impedance measurements to derive the contributions from the various cellular barrier compartments. Ion current flow between the basal cell layer and the substrate can be described by an indicative parameter Alpha (α) [4]. However, due to the multifrequency nature of ECIS, the values Rb, α, and Cm can be mathematically modeled to track the osteogenic differentiation process. To distinguish between proliferating and differentiating stem cells, a detailed analysis may be favorable, not only on the impedance parameters but also on the cytoskeletal and adhesion-relevant properties of the cells. Our present study combines the electrical stimulation modalities of titanium implants with the ability to follow the osteogenic differentiation process of the stem cells in real time by ECIS. This opens the possibility of developing an entirely new type of implant that can induce osteogenesis and permit monitoring the induction process. We aim to integrate the electrical stimulation with the osteogenic response: the speaking implant.

## 2. Materials and Methods

### 2.1. Cell Culture

Adipose-derived stem cells (ASCs) were isolated from lipoaspirate samples collected from patients who have undergone liposuction or lipofilling procedures at the Department of Oral and Maxillofacial Surgery, Facial Plastic Surgery, Rostock University Medical Center, Germany, after approval from the local Ethics Committee (No. A 2014-0092). This procedure has been described before [8,21]. Briefly, the samples were washed with PBS and digested with 6 mg/mL collagenase NB4 (SERVA Electrophoresis GmbH, Heidelberg, Germany) for 30 min. After filtration through a 100 μm cell strainer (Becton Dickinson, Franklin Lakes, NJ, USA), 10% Hyclone Newborn Calf Serum (NCS; Sigma Aldrich Chemie GmbH, Munich, Germany) was added and the sample was centrifuged at 1000 rpm for 10 min. The pellet was washed in 10 mL PBS/10% NCS, centrifuged again, and resuspended in 10 mL PBS/10% NCS. Stem cells were expanded on cell culture flasks harboring specialized hydrophilic surfaces (Sarstedt, Germany). Stemness was identified according to the positive expression of the surface markers CD13, CD73, CD90, and CD105. ASCs from the following donors were used (Table 1).

Human mesenchymal stem cells isolated from bone marrow (BMSC) as well as primary osteoblasts (POB) were purchased from Promocell (#C-12974, # C-12720; Heidelberg, Germany). ASCs and BMSCs were cultured in PromoCell MSC Growth Media (C-28009, Promocell, Heidelberg, Germany) at 37 °C in 5% CO_2_ and a 95% air-humidified incubator. POBs were grown in Dulbecco’s modified Eagle’s medium (Invitrogen, Karlsruhe, Germany) with 10% fetal bovine serum (PAN Biotech GmbH, Aidenbach, Germany) and 1% gentamycin (Ratiopharm, Ulm, Germany). Stem cell characterization was performed as described before [8]. 

### 2.2. Impedance Sensing and Calculation

Impedance was measured with a commercially available Electric Cell-Substrate Impedance Sensing system (ECIS Zθ; Applied BioPhysics, Troy, NY, USA) equipped with a 96-well array station (AppliedBiophysics, NY, USA) to monitor time (t) and frequency (f) as well as dependent complex impedance (Z(t,f)) of proliferating and differentiating cells. ASCs, BMSCs, and POBs were grown on 96-well ECIS array plates with 20 interdigitated electrodes per well (96W20idf PET; ibidi GmbH, Gräfelfing, Germany). Before cell seeding, electrodes were stabilized with serum-free media overnight in the incubator with high humidity at 37 °C and 5% CO_2_. Cells were seeded directly on the electrodes at a cell density of 10,000 cells per well. After 24 h of adhesion, impedance sensing was started. The differentiation process was initiated after 48 h of cell seeding:Group 1 (control): chemical induction of osteogenic differentiation in ASC, BMSC, and POB;Group 2a: continuous electrical stimulation for three or seven days in ASC, BMSC, and POB;Group 2b: pulsatile electrical stimulation for three or seven days in ASC, BMSC, and POB.

All measurements were performed in the appropriate cell culture medium by real-time monitoring of impedance alterations at 11 frequencies (0.0625, 0.125, 0.25, 0.5, 1, 2, 4, 8, 16, 32, and 64 kHz) in a 180 s interval following the ECIS Handbook for a minimum of seven days. Calculation was performed using the model from Giaver and Keese [4], which is directly implemented in the ECIS Zθ software. Out of this modeling, three time-course frequency-independent impedance relevant parameters were calculated: Rb (ohm·cm^2^), resistance with respect to the cell-covered area between cells with more or less tight cell-to-cell junctions;α (Ω^0.5^.cm), describing the cell-to-substrate characteristics;C_m_ (μF/cm^2^), the average cell membrane capacitance, attributed to charge separation along phospholipid boundaries of the cellular membrane.

### 2.3. Chemical Osteogenic Differentiation

Osteogenic differentiation was initiated by changing the growth medium to the Mesenchymal Stem Cell Osteogenic Differentiation Medium (C-28013, Promocell, Heidelberg, Germany) in ASC, BMSC, and POB of Group 1 (control). Alizarin Red staining was performed as described previously to confirm osteogenic differentiation in the control and test groups [8]. For this purpose, alizarin fluorescence was visualized by Axio Scope.A1 fluorescence microscope (Carl Zeiss, Jena, Germany) using AxioVision Imaging Software 4.8.2.0 (Carl Zeiss, Jena, Germany). Osteogenic marker expression was determined by RT-PCR using Primer against the genes of osteopontin (OPN) and bone morphogenetic protein 2 (BMP-2) [22].

### 2.4. Electrical Stimulation

The electrical stimulation chamber was built by rapid prototyping technology (IPT, Wismar, Germany) and is composed of Foto Med LED.A material (Innovation MediTech GmbH, Unna, Germany) [8,23]. Two triangular electrodes with a length of 23 mm are composed of Ti6Al4V and are connected by a polyether ether ketone (PEEK) isolator, visible as a light yellow centerpiece (Figure 1A,B). The current was applied to the respective outer end of the titanium electrodes.

All stimulation experiments were performed with ASCs, BMSCs, and POBs seeded on both electrodes with a density of 3.0 × 10^4^ cells/cm^2^ and on coverslips with a thickness of 2.2 × 10^4^ cells/cm^2^. After 24 h of cell adhesion and spreading, electrical stimulation was started with 0.7 V AC/20 Hz for three or seven days in a conventional incubator at 37 °C under 5% CO_2_ atmosphere by connecting the outer ends of the titanium electrodes to the Metric GX 305 function generator (Metrix Electronics, Hampshire, UK) set to 20 Hz. For continuous stimulation, the applied AC signal was never interrupted. In contrast, for the pulsatile inspiration, the electric field was applied three times at 45 min each day. Note that the 45 min stimulation period was evenly distributed over the day (Figure 1C). 

### 2.5. Microscopy Techniques

#### 2.5.1. Scanning Electron Microscopy

Electrodes with seeded cells that were electrically stimulated for up to three days were analyzed via field emission scanning electron microscope (FE-SEM, MERLIN^®^ VP Compact) equipped with an energy dispersive X-ray (EDX) detector (XFlash 6/30) and analysis software Quantax Esprit 2.0, (Co. Bruker, Berlin, Germany). Representative areas of the samples were analyzed and mapped to determine the elemental distribution on the basis of the EDX-spectra data by the Quantax Esprit Microanalysis software (version 2.0). In brief, cells were washed with PBS, fixed with 2.5% glutardialdehyde (Merck) for one h at room temperature, and stored until microscopy at 4 °C. After washing with 0.1 M Na-phosphate buffer, the samples were dehydrated in an ascending series of ethanol and were critical point dried (Emitech K850, Co. Quorum Technologies LTD, East Sussex, UK). Samples were mounted on Aluminum SEM carrier with adhesive conductive carbon tape (PLANO, Wetzlar, Germany) and coated with carbon under vacuum (EM SCD 500, Co. Leica, Bensheim, Germany; Figure 1D,E).

#### 2.5.2. Confocal Laser Scanning Microscopy

Our group already described the general confocal microscopy procedure [24]. For this purpose, continuous and pulsatile electrically stimulated (Group 2a/b either for 3 or 7 days stimulation) as well as non-stimulated ASC, BMSC, and POB cells were fixed in 4% paraformaldehyde (Santa Cruz, Dallas, TX, USA), permeabilized with 0.1% Triton X-100 (Santa Cruz, Dallas, USA) and labeled with Phalloidin-Alexa 488 (Invitrogen) or focal adhesion kinase (FAK) primary antibody (#3285, Cell Signaling, USA) combined Alexa488 secondary antibody (Cell Signaling, Danvers, MA, USA) and counter-stained with Hoechst (PanReacAppliChem, Darmstadt, Germany). Microscopy was performed with an inverted confocal laser-scanning microscope (LSM780, Carl Zeiss, Jena, Germany). All images were taken at identical device settings to guarantee comparable results. The image processing was carried out using ZEN 2011 (Carl Zeiss Jena GmbH, Jena, Germany).

### 2.6. Initial Adhesion Measurements

ASCs were electrically stimulated for three or seven days (Group 2a/b) and detached by Trypsin EDTA afterward. Cells were counted and seeded in three different concentrations of 1000, 5000, and 10,000 cells in 12-well plates. After 1, 2.5, and 24 h initial adhesion time, the process was stopped by washing with ice-cold PBS and adding ice-cold methanol for 10 min. Methanol was aspirated from the plates. Then, 0.5% crystal violet solution (Sigma, Germany) diluted in 25% methanol was added to the cells and incubated for 10 min. Cells were washed several times with water until no dye was excreted anymore. Plates were dried by air. Crystal violet-stained ASC cells were measured at 590 nm in a conventional microplate reader (TECAN Infinite^®^ 200 PRO, Tecan, Männedorf, Switzerland).

### 2.7. Statistics

All experiments were replicated at least three times with individually passaged cells, and data sets were expressed as means ± standard deviations (SD) using the software GraphPad Prism Version 5 (GraphPad Software, La Jolla, CA, USA, accessed on 23 March 2011) or MS Excel 2020. Normal distribution was checked using the non-parametric Kolmogorov–Smirnov test, and results were analyzed for statistical significance using analysis of variance (ANOVA), unpaired non-parametric Mann–Whitney U tests, Wilcoxon–Whitney tests, and Students’ *t*-test. The level of statistical significance was set to *p* ≤ 0.05.

## 3. Results

### 3.1. The Electrical Stimulation System

Adipose-derived stem cells (ASCs), bone-marrow-derived stem cells (BMSC), and primary osteoblasts (POB) were stimulated on two corundum-blasted Ti6Al4V electrodes (Figure 1A,B), either under continuous or under pulsatile (3 × 45 min per day) current conditions for three or seven days at 0.7 V AC/20 Hz. Surface roughness and element analysis of the electrodes, including spread cells on top after electrostimulation, were analyzed by scanning electron microscopy (Figure 1D,E; exemplarily ASCs are presented after continuous stimulation for three days). The element titanium, as one of the material components of the electrode, is shown in green. In Figure 1D, the aluminum of the electrode alloy is marked in red. The cells spanning the surface are presented in blue by carbon labeling. Figure 1E shows a more central part of the electrode, over which cells have spread very well, which shows their fibroblast-like morphology. The silicon dioxide particles left by sandblasting can be recognized by their pink color.

### 3.2. Impedance Sensing after Chemical Osteogenic Induction

To get a first impression of the impedance alterations under osteogenic differentiation, ASC, BMSC, and POB cells were chemically osteogenic differentiated by changing the culture medium to an osteogenic medium. Here, all cell types were seeded onto the ECIS electrodes, and after an initial adhesion phase of two days, osteogenic induction was started. To monitor the effects of the impedance measurements at different frequencies, the resistance R_b_ was measured at 4, 16 kHz and the electrical capacitance C_m_ at 64 kHz (Figure 2). At a low frequency of 4 kHz, the resistance of the cell-covered electrodes displayed fundamental values of 250 to 300 Ω at the seeding point (t = 0 h). Over time, the resistance values increased due to cell attachment, spreading, and proliferation. However, after initiating the osteogenic differentiation at the 48 h time point, the resistance of BMSC and ASC cultures increased more strongly to values of 450–500 Ω during the first four to five days before decreasing again.

In comparison, the impedance of POB cells did increase at a slower rate until it remained relatively stable after day five (Figure 2A). A frequency of 16 kHz showed similar average impedance rates, referring to a mean value that indicated the resistance that passes through the cell membranes, the cell monolayer, and the cell distance to the substrate. This impedance showed a significant increase from 250 Ω to 450 Ω after osteogenic differentiation initiation in ASC and BMSC in contrast to the POB cells (*p* < 0.01; time points 48–96 h) (Figure 2B). Notably, ASC showed the steepest rise in impedance of all analyzed cells. Medium changes at the 96 h time point resulted in a continuous decrease in the impedance values, except for POB. POBs’ impedance values were continuously increasing regardless of whether osteogenesis was induced or the medium was changed, indicating no response to the osteogenic differentiation medium. A frequency of 16 kHz was observed to analyze the monolayer’s capacitance, which is most suited to follow the increasing surface coverage of the electrodes by cells. An inverse correlation to the resistance data at 4 kHz could be observed here. For ASC and BMSC cultures, the capacitance decreased from 60 to 20 nF after osteogenic differentiation induction. In contrast, the capacitance of the POBs continuously decreased after seeding (Figure 2C). Overall, these results demonstrate that the best discrimination by impedance measurements between the onset of osteogenic differentiation of stem cells and moderate proliferative effects of primary osteoblasts can be achieved at 16 kHz.

### 3.3. Continuous versus Pulsatile Electrical Stimulation for Three Days

Based on the results mentioned above, a frequency of 16 kHz was selected based on the description of the differentiation via electrical stimulation. The respective impedance was recorded three days after either permanent or pulsatile electric stimulation of ASC, BMSC, and POB (Figure 3). After seeding (t = 0 h), the starting impedance values varied between 280–320 Ω. For the stem cells (ASC, BMSC) under control conditions (i.e., without electrical stimulation and chemical osteogenic differentiation), the impedance showed no significant fluctuations up or down, as evident from Figure 2 before osteogenic differentiation. A completely different picture emerged after the electrical stimulation of the stem cells. Both the continuous and the pulsatile stimulated stem cells (ASC, BMSC) showed an almost linear increase in impedance. The permanent electrical stimulation caused a significant increase (280–440 Ω) compared to the pulsatile stimulation (280–380 Ω). Nevertheless, the high impedance values of 500 Ω, displayed after chemical osteogenic induction (Figure 2), could not be reached after electrical stimulation for three days. In contrast, the electrically stimulated primary osteoblasts (POB) behaved like the non-stimulated control cells, regardless of the type of stimulation protocol. A continuously, linearly increasing change in impedance became evident, indicating a continuous growth of the POBs (starting values from 300 Ω to 400 Ω). These data also confirm the measurements from Figure 2, in which the POB were chemically differentiated. Data sets concerning the measures of the resistance at 4 kHz and capacitance at 64 kHz compared to the impedance at 16 kHz are presented in Appendix A.

As an interim conclusion, it can be summarized that firstly, the stem cells from fat (ASC) and bone (BMSC) react equally to chemical and electrical stimulation; secondly, the stem cells permanently stimulated for three days showed a more significant increase in impedance compared to those which were pulsatile stimulated. However, we recently showed that short periods of permanent electrical stimulation led to increased proliferation of stem cells [8]. Therefore, distinguishing between proliferation and differentiation with similarly increasing impedance values arises. To answer this question, the first approach was to increase the electrical stimulation durations by up to seven days because we recently demonstrated that longer permanent stimulations lead to growth arrest and apoptosis initiation.

### 3.4. Continuous versus Pulsatile Electrical Stimulation for Seven Days

As ASC and BMSC have shown to behave in the same way under electrical stimulation conditions and the POB did not respond to the stimulation, the following experiments were carried out with ASC only. For seven days of either pulsatile or permanent electrical stimulation, the impedance at 16 kHz was monitored (Figure 4A). Control ASC showed slowly increasing impedance values due to the slowly increasing proliferation (220–230 Ω). Similarly, permanently stimulated ASCs displayed no significant impedance alterations within the week of recording (remained at 220 Ω). In contrast, pulsatile stimulated ASCs showed continuously rising impedance values, starting at 220 Ω and increasing up to 280 Ω, similar to those results from chemical induction (Figure 2B) and the 3-day pulsatile stimulated ASCs (Figure 3). The impedance levels off shortly to 260 Ω before increasing again after a medium change. (The onset of this saturation can also be found in the data shown in Figure 3.)

To verify if seven days of pulsatile electrical stimulation induced osteogenesis, calcification was proven by Alizarin Red staining after 21 days (Figure 4B). The red-stained calcium deposits are visible on the pulsatile stimulated ASCs. In contrast, no red color could be seen in the continuously stimulated cells. In addition, the expression of two critical osteogenic markers was detected: osteopontin (OPN) and bone morphogenetic protein 2 (BMP-2) (Figure 4C). As a positive control (P-Ctrl), chemically osteogenic stimulated stem cells were used. The negative control cells (Ctrl) were maintained in an expansion medium. Pulsatile electrically stimulated ASCs (7 d) showed a significantly increased expression of OPN and BMP2.

Table 2 Differences in the impedance values (Ω) of the stem cells of adipose origin (ASC), bone marrow (BMSC), and primary osteoblasts (POB) measured at 16 kHz under control conditions, chemically osteogenic stimulated or electrically stimulated for 3 or 7 days (d), either continuous or pulsatile. Significantly different values compared to the respective control are highlighted with * (mean ± SD, n = 3–5, *p* < 0.05). n.d., not detected. To distinguish pure proliferation from osteogenic differentiation in all stimulated cells, we applied the model developed by Giaever and Keese [3,4] to interpret changes of impedance over time described by Z(t,f) during early induction for three impedance-relevant parameters (Figure 5A). First, α, a parameter related to the cell-to-substrate interactions was calculated for chemical osteogenic induced ASCs and BMSCs; a significant increase from α = 1.4 to 2.4 Ω^0.5^.cm was observed (Figure 5B). In comparison, for seven days of pulsatile electrical stimulation, ASCs also showed elevated α-values starting at time point 140 h from α = 1.4 to 1.6 Ω^0.5^.cm. No alterations for α were observed for the three POB groups, control, and electrically continuously stimulated ASCs and BMSCs. The barrier resistance, R_b_, which describes the cell layer and the establishment of tight junctions, was strongly elevated in osteogenic differentiated ASCs and BMSCs (R_b_ = 0.8 Ωcm^2^) and slightly increased in seven days pulsatile stimulated ASCs (R_b_ = 0.1 Ωcm^2^) (Appendix A). The third parameter, the cell membrane capacitance, C_m_, revealed a continuous decrease for the osteogenic stimulated ASCs and BMSCs (C_m_ = from 2.0 to 0.5 μF/cm^2^), followed by the pulsatile electrically stimulated ASCs (C_m_ = from 1.5 to 0.1 μF/cm^2^). Astonishingly, we could not detect any changes in C_m_ for any of the cells after three days of stimulation (Appendix A).

### 3.5. Structural, Morphological, and Adhesion-Related Alterations

Alterations in the three impedance-based parameters, specifically in α, indicate morphological alterations of the cells, structural modifications of the barrier function, or modulation of the cell membrane integrity. Therefore, a first overview of the cell shape and cytoskeleton formation was taken by labeling the F-actin fibers (Figure 6A). Most noticeably, after three days of pulsatile electrical stimulation, an increased actin stress fiber formation was seen in all three cell types. These stress fibers, which span the entire cell, lead to cell stabilization, making the cells appear more spread out and stiff. However, permanent BMSC stimulation increased the formation of cell protrusions, which were not observed for ASCs and POBs (Appendix A).

Besides these morphology modulations, cell membrane-associated proteins and structures were monitored. The cell structure, cell–cell contacts, and membrane properties were examined by scanning electron microscopy. Confluent ASCs showed no apparent differences after control treatment or electrical stimulation. However, enlarged images of subconfluent cells showed a decrease in microvilli-like structures on the cell surface after electrical stimulation (Figure 6A, lower row). Pulsatile stimulated ASCs showed a reduced number of these structures, while continuously stimulated ASCs produced almost no more microvilli filaments. The cell surface looked polished. In the next step, several adhesion-related proteins were labeled to analyze the cell-to-substrate or cell-to-cell modifications.

Exemplarily, the focal adhesion kinase (FAK), a trans-membrane protein involved in cellular adhesion and spreading processes, was labeled in control-treated, pulsatile, or continuously stimulated ASCs (Figure 6A, middle row). The control ASCs expressed FAK in the cytosol, from where it is transported to the cell periphery and, in cooperation with other proteins, controls the adhesion machinery. FAK clusters in the cell periphery are indicated with white arrows and are evenly distributed over the entire surface. In contrast, pulsatile stimulated cells showed only very few FAK clusters in the cell periphery with a dominance of strongly pronounced stress fibers.

In contrast, the continuously produced cells showed an increased peripheral FAK expression. Many FAK clusters lined up on the cell surface and thus mediated a stable connection between the cell and the extracellular matrix. To prove our hypothesis that pulsatile stimulation stabilizes the cell and continuous stimulation modulates the adhesion properties, an initial adhesion assay was performed (Figure 6B). The results demonstrate a significant decrease in initial adhesion after pulsatile stimulation. In brief, about 40% fewer ASCs adhered to the substrate compared to the non-stimulated control. In contrast, continuous stimulation caused a significant increase in cellular adherence of 10%. Figure 7 summarizes the results that were obtained after pulsatile and constant stimulation. The pulsatile stimulation resulted in decreased space between the cellular membrane and the substrate accompanied by increasing α values. It appears that the cells nestle against the ground, stiffening the actin cytoskeleton and decreasing the number of basal adhesion proteins. At the same time, the number of cell-to-cell connections increased slightly, so the R_b_ value increased somewhat. These morphological alterations are indicators of the initiated osteogenesis.

In contrast, the continuous stimulation caused the cells to round off, accompanied by increased expression of the adhesion molecules in the periphery to prevent the cells from detaching. The actin fibers are mainly distributed to the cell periphery. Further, the tight junctions between the cells become rare, so the stem cells lose contact with the substrate and the neighbor cells. A condition that ultimately may lead to apoptosis induction.

## 4. Discussion

One way to overcome xenogenous, alloplastic, allogeneic, and autologous grafting procedures in the maxillofacial area is the electrical stimulation of stem cells. This approach could prevent the risks and limitations of grafting [25,26,27]. In a previous study, our group evaluated the electrical stimulation parameters to improve human adipose-derived stem cells (ASCs) proliferation in vitro [8]. Furthermore, these optimal stimulation parameters have been calculated by in silico models, and the electric field distribution was modeled by finite-element simulations [28]. The settings of 1.7 V AC/20 Hz under continuous conditions revealed a proliferation increase within the first three days, whereas more extended stimulation periods forced metabolic arrest and apoptosis initiation.

In contrast, a pulsatile stimulation was applied with periods of 3 × 45 min per day with a 225 min break between the stimuli on primary osteoblasts, which resulted in enhanced osteocalcin and collagen I expression. Thus, we decided to compare the continuous and pulsatile electrical stimulation on stem cells of adipose (ASC) and bone marrow (BMSC) origin as well as on primary osteoblasts (POB) by impedance monitoring. In addition, the electrical stimulation results were compared to the chemically induced osteogenic differentiation of the respective cells. First, ASCs and BMSCs are especially suitable as they are abundant, accessible, and functional with an osteogenic potential [29,30]. Secondly, modifications in the electric impulse parameters are known to modulate different cellular signaling pathways [31,32,33]. Third, impedance measurements are suitable for osteogenesis monitoring [6,7]. 

After we had verified that the cells adhered well to the electrodes and spread uniformly over the rough surface, we chemically induced the osteogenic differentiation and compared the impedance measurement data with those from the literature. A strong increase in the impedance values (difference to control: ~200–400 Ω), regardless of whether recorded at a frequency of 16 (our measurement), 40, or 64 kHz, marks the osteogenic initiation process [6,7]. A strong increase at 16 kHz could also be observed after three-day continuous (difference to control: ~100 Ω), pulsatile (difference to control: ~75 Ω), and seven-day pulsatile (difference to control: ~55 Ω), electrical stimulation of the ASCs, which indicates that osteogenesis might also be induced under these conditions. However, continuous electrical stimulation over seven days did not alter the impedance values. This was to be expected because our previous work impressively showed that this continuous stimulation leads to a growth arrest and an apoptosis initiation after seven days [8]. However, similar results emerged here for both continuously and pulsatile stimulated cells, regardless of whether cells of adipose or bone-marrow origin were used: the impedance continuously rose over time. The impedance data could be interpreted so that osteogenic differentiation is initiated in all cells treated electrically for a minimum of three days. However, this is not the case. Our previous work demonstrated that continuous electrical stimulation leads to a 4.5-fold increase in the cell number and a significant 5-fold higher arrest in the G2/M cell cycle phase of the stem cells, which can be explained by cellular activation via asymmetric redistribution/diffusion of electrically charged cell membrane receptors in response to electric fields and via cell membrane depolarization due to direct activation of voltage-gated Ca^2+^ channels [34]. Based on these results, the question arises of how and whether it is possible to differentiate between proliferation and osteogenesis using impedance measurements. For this purpose, the impedance data were disassembled according to the ECIS model developed by Giaever and Keese. It is known that a gradual reorganization of the cell layer occurred throughout differentiation [4]. Here, the frequency-dependent complex impedance, Z(t,f), was dismantled in the three parameters: Alpha (α), C_m_, and R_b_. Given a confluent monolayer, α can be the parameter describing the cell–matrix interaction, as shown for Madin–Darby canine kidney cell line (MDCK) [35]. So far, reported α-values vary between 16 ± 0.6 Ω^0.5^.cm for MDCK-II cells, 5.3 ± 0.3 Ω^0.5^.cm for HUVECs, and 7 Ω^0.5^.cm for lung fibroblasts [35,36]. Bagnaninchi and Drummond measured relatively high levels of α = 5.25 Ω^0.5^.cm within the first days of osteogenic differentiation in ASCs of not featured origin [6]. In accordance with this, the α-values of the stem cells of adipose and bone-marrow origin used in our present study also displayed an increase in α from 1.4 to 2^0.5^.cm after switching to the differentiation medium. The observed values are lower than those described in the literature. One reason could be the age of the ASC donors, which may temporally control the onset of osteogenesis. Nordberg et al. reported that ASCs from younger donors require more time to differentiate than ASCs from older donors. Still, young ASCs proliferated more and accreted more calcium in the long term [7]. Here, the initial α-values also varied depending on the age and the medium (young donors about 3.0 Ω^0.5^.cm, old donors about 5.5 Ω^0.5^.cm). However, the increasing α-values indicate the onset of osteogenic differentiation, ultimately also because the control cells and strongly proliferating cells showed no increase in α. Furthermore, the α-value correlates with the distance between the cell and the electrode. Increasing α-values mean that the distance of the cell to the substrate becomes smaller. This is because the transformation of electrical stimulation into mechanical activity induces tension in the cytoskeleton via the reorganization of cytoskeletal filaments and redistribution of actin and membrane proteins [9,37]. Additionally, after pulsatile electric stimulation, the actin stress fibers were reinforced—long, stable fibers pervaded the entire cell. In conjunction with stabilizing the actin cytoskeleton, some adhesion-associated proteins are modulated in their expression and distribution. For example, pulsatile stimulated ASCs exhibit a significantly reduced expression of the focal adhesion kinase in the cell periphery indicating altered membrane-associated cellular adhesion processes. However, this actin-mediated cell stabilization and paralyzed adhesion machinery were proven by significantly lowered initial cell adhesion. Pulsatile electrically stimulated ASCs stay stiff even after detachment from the substrate, so re-attachment is impeded. The rearrangement of the focal adhesion kinase additionally complicates the renewed initial adhesion. Indeed, some previous studies showed a connection between osteogenesis and the activation of focal adhesion kinase in combination with ERK1/2-RUNX2 co-activation [37,38]. Though different, only the focal adhesion kinase expression levels were examined in these studies, not their intracellular distribution or reorganization.

## 5. Limitations and Future Directions

In addition to increasing the number of biological replicates, possibilities for optimizing this in vitro study include extending the experiments to include other differentiation processes in human stem cells. Besides initiating or accelerating osteogenesis, chondrogenesis could also be another starting point for electrically inducible implant systems. However, the most challenging of this in vitro preparatory work is the transfer to an animal model. Currently, we are facing the development of electrically stimulable electrodes into Aachener Minipigs. The electrodes and implantable stimulation system were designed in correspondence with the in silico modeling of an alloplastic, electronically active bridging system (osteosynthesis) for critical size defects of the mandible. After establishing the implant system, the second step is to read out the impedance values in the animal model. This poses another challenge because modeling the Alpha requires measuring the impedance values in a large frequency spectrum. 

## 6. Conclusions

Our present study confirms that impedance measurements are suitable for monitoring differentiation processes by reading out the parameter in conjunction with literature data. α describes the cell–matrix interaction, and the increase in its value indicates the onset of osteogenic differentiation. These measurements are not limited to stem cells, which are chemically differentiated but can also be transferred to other differentiation systems. To our knowledge, this study is the first to describe the combination of the electrical stimulation and differentiation of several types of stem cells by multimodal impedance sensing. Finally, taken together, modeling α helps to distinguish between proliferating and differentiating stem cells. However, we recommend that for future work on real-time label-free monitoring of osteogenic differentiation processes, the impedance measurements should be supplemented by assessing the cell stiffness and possibly the distribution of defined adhesion molecules. For this purpose, the development of specific sensor chips is in progress. Furthermore, our results open the possibility of developing entirely new implants that can induce osteogenesis and follow its process.

## Figures and Tables

**Figure 1 biomedicines-11-00697-f001:**
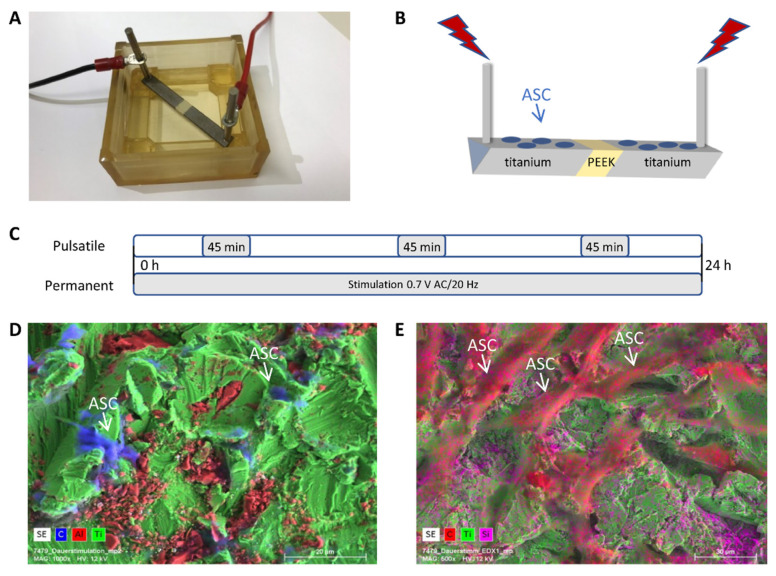
The electrical stimulation system, its components, and the electrode surface. (**A**) The electrical stimulation chamber for both continuous and pulsatile stimulations. (**B**) Schematic presentation of the two electrodes connected with a PEEK isolator in the middle. Cells were seeded on the top of the electrodes. Stimulation parameters (0.7 V AC/20Hz) can be applied continuously or pulsatile (3 × 45 min/d) for 3 or 7 days at 0.7 V AC/20 Hz. (**C**) Schematic representation of the differences between pulsatile and continuous stimulation. (**D**,**E**) Scanning electron images include EDX element analysis for titanium (green), aluminum (red), silicon (pink), and carbon (blue).

**Figure 2 biomedicines-11-00697-f002:**
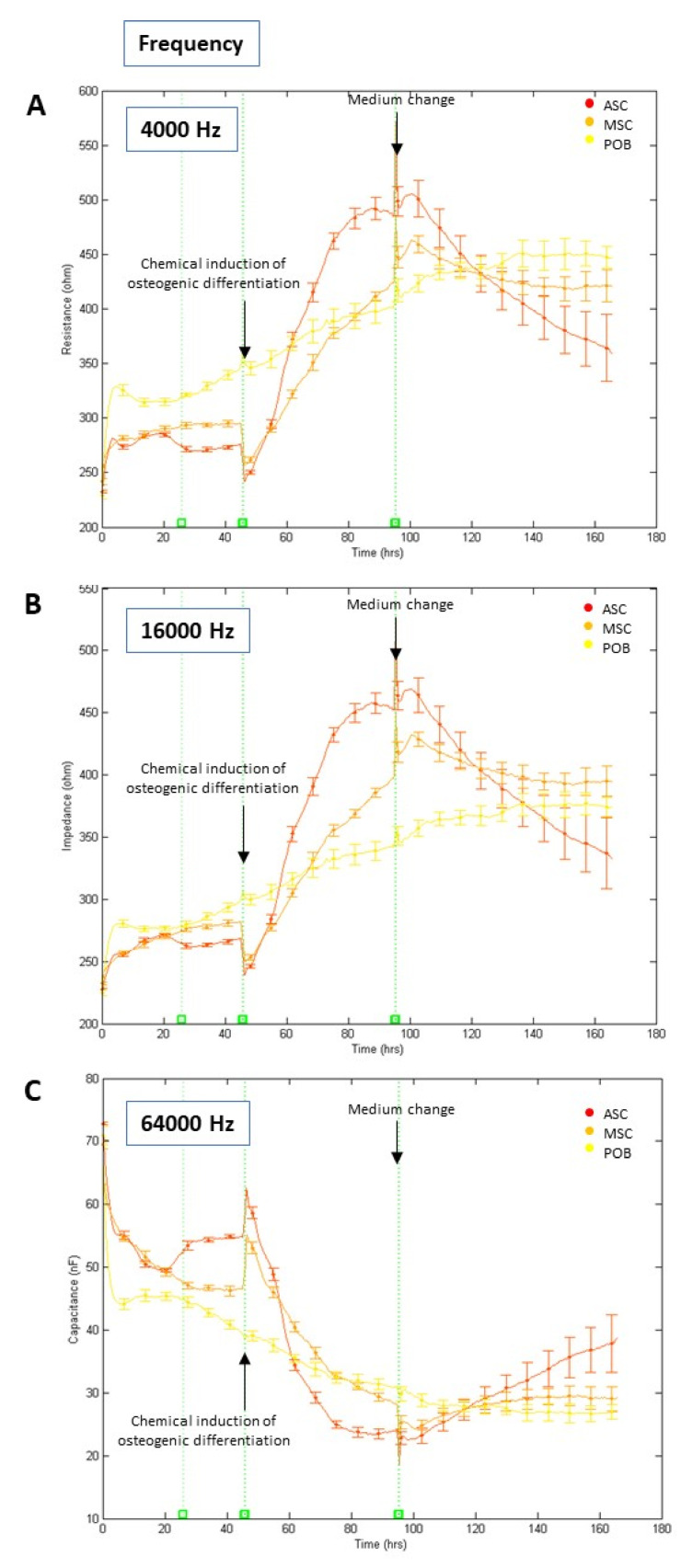
(**A–C**) Determination of impedance parameters after chemical osteogenic induction. Normalized real-time and label-free impedance (frequencies at 4 (2A), 16 (2B), 64 (2C) kHz) curves of human adipose-derived stem cells (ASC, red line), bone-marrow-derived stem cells (BMSC, orange line), and primary osteoblasts (POB, yellow line), (mean ± SD, n = 3). All cells were seeded (t = 0 h) in multi-well electrode arrays. Chemical osteogenic induction was started (t = 48 h) by supplementation with an osteogenic differentiation medium. At t = 96 h, the last medium change was carried out. The total measurement time was seven days. Note that only the stem cells responded with a substantial impedance increase. At the same time, most of the already fully differentiated osteoblasts showed no significant impedance alteration after supplementation with the osteogenic differentiation medium. Further, medium changes caused short-term impedance alterations due to the small temperature fluctuations.

**Figure 3 biomedicines-11-00697-f003:**
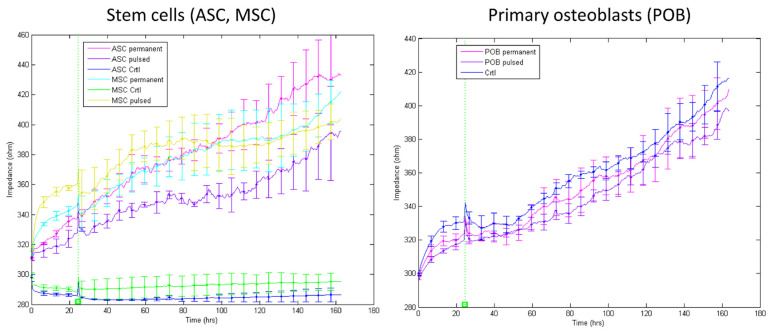
Impedance measurements at 16 kHz after three days of electrical stimulation. Impedance was measured in non-stimulated, pulsatile, and continuously electrically stimulated ASCs, BMSCs (**left**), and POBs (**right**). At t = 0 h, cells were seeded on the chips. Medium changes occurred every two days (mean ± SD, n = 3–5).

**Figure 4 biomedicines-11-00697-f004:**
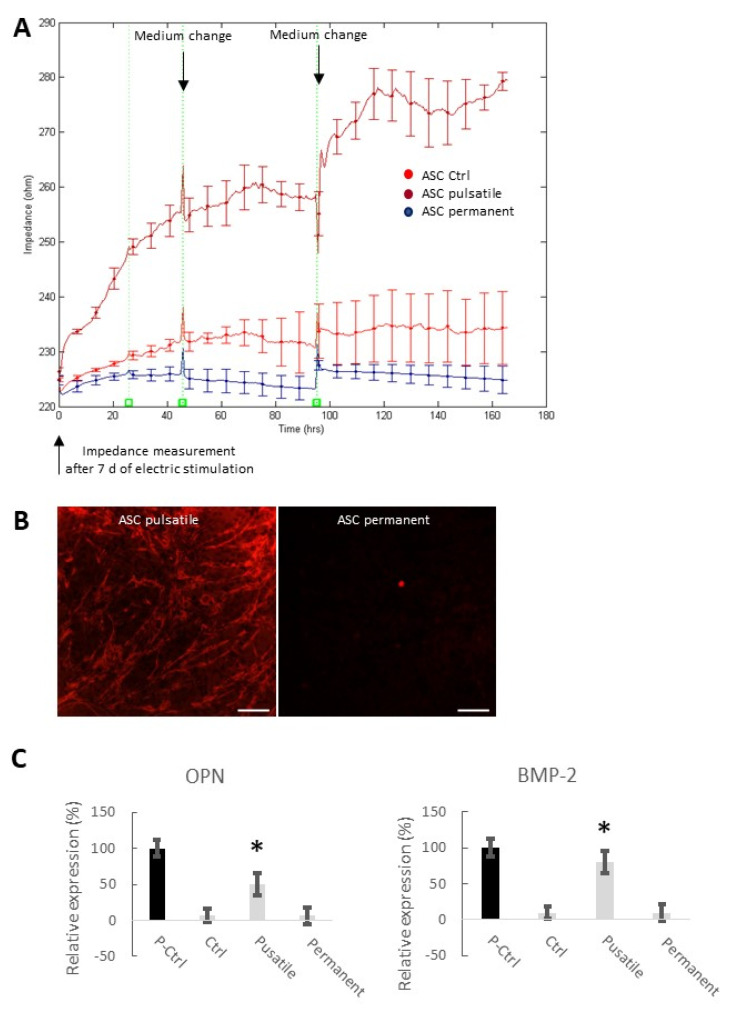
Impedance measurements at 16 kHz after seven days of electrical stimulation. (**A**) Impedance was measured in non-stimulated, pulsatile, and continuous electrically stimulated ASCs. At t = 0 h, cells were seeded on the chips. Medium changes occurred every two days (mean ± SD, n = 3–5). (**B**) Alizarin Red staining of the calcium deposits in pulsatile and continuous electrically stimulated ASCs after 21 days. Note that medium changes caused short-term impedance alterations due to the small temperature fluctuations. Scale bar: 100 µm. (**C**) Expression analysis of two relevant osteogenic marker genes (i) osteopontin (OPN), also known as bone sialoprotein I, and (ii) bone morphogenetic protein 2 (BMP-2). Notably, transcript levels of the two marker genes of the positive control (P-Ctrl; chemically differentiated ASCs) and the electrically, pulsatile stimulated ASCs for seven days are significantly higher in comparison to the negative control (Ctrl) and the permanently stimulated stem cells (mean ± SD, n = 3; * *p* < 0.05 considered significantly different compared to the negative control (Ctrl)).

**Figure 5 biomedicines-11-00697-f005:**
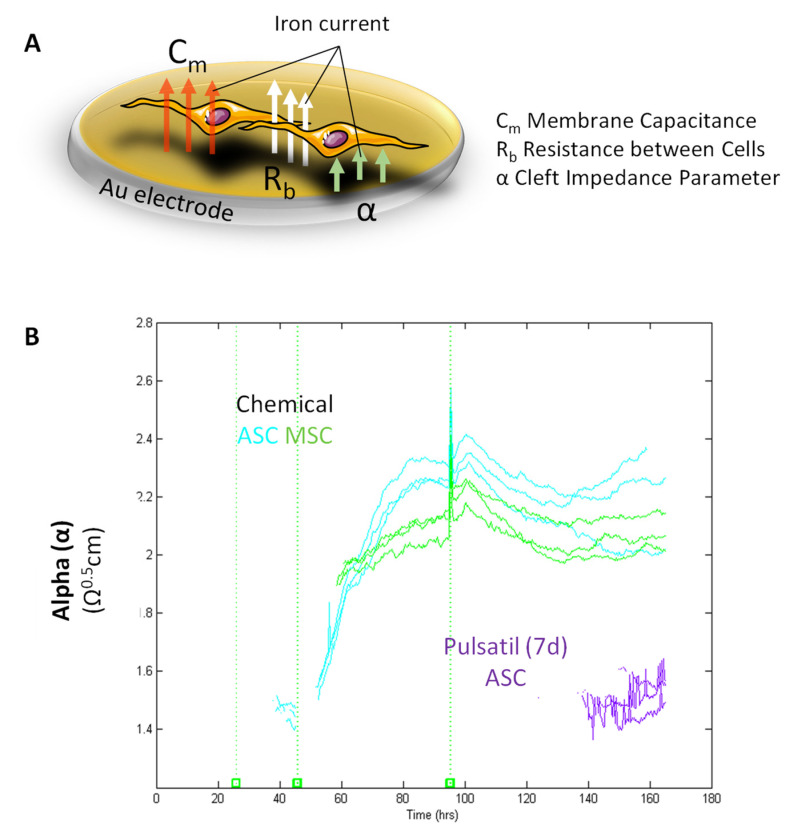
Impedance Modeling. (**A**) Schematic illustration presenting the modeled parameters R_b_, α, and C_m_. R_b_ describes the paracellular barrier between cells caused by tight cell-to-cell junctions; α models the basal adhesion, indicating the cell-to-substrate interaction; C_m_ shows the cell membrane capacitance. Notably, α is dependent on the cell radius and the subcellular adhesion. Adapted by Giaver and Keese [4]. Created with www.biorender.com. (**B**) Calculation of the cell-to-substrate interactions (α) in chemical osteogenic and electrically stimulated stem cells from the adipose origin (ASC) and bone marrow (BMSC). The equation for calculating α is given in the right corner: r_c_ = cell radius, p = specific electrolyte resistivity in the cleft, and h = distance between membrane and electrode. Notably, the higher the distance (h) from cells to substrate, the smaller α. Chemical osteogenic stimulated ASCs and BMSCs revealed a continuous increase in α over time. Pulsatile electrical stimulation of ACSs also showed a rise in α starting at 140 h. During seven days, no alterations in α were observed for POBs, control-treated, and electrically continuously stimulated stem cells (n = 3–5).

**Figure 6 biomedicines-11-00697-f006:**
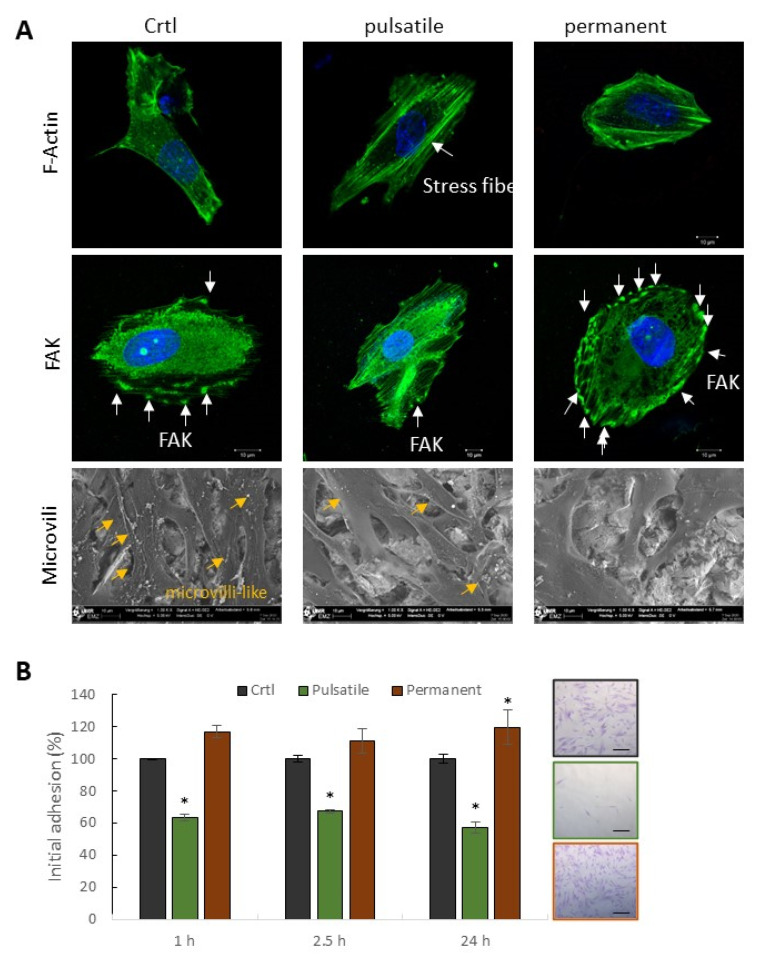
Morphological and adhesin-related alterations. (**A**) Actin cytoskeleton (F-actin, upper row), focal adhesion kinase (FAK, middle row), and microvilli (lower row) visualization after three days of pulsatile or continuous electrical stimulation in comparison to non-stimulated ASCs. After pulsatile stimulation, a boosted actin stress fiber formation (green) was detectable. FAK molecules are significantly reduced after pulsatile stimulation, while continuous stimulation increases the amount of FAK in the cell periphery. F-actin and FAK were counterstained with Hoechst in blue to mark the nuclei. Scanning electron images indicate the microvilli-like structures on the cell surface, which are reduced in pulsatile stimulated cells and seem largely absent in continuously stimulated cells. (**B**) Comparative analyses of initial adhesion of non-stimulated (control), pulsatile, and continuous electrically stimulated ASCs after three days. Three adhesion time points (1, 2.5, 24 h) were chosen to monitor the adhesion process. Right: Exemplarily pictures of adherent cells after one h. Mean ± SD, n = 3, * *p* < 0.05. Scale bar: 100 µm.

**Figure 7 biomedicines-11-00697-f007:**
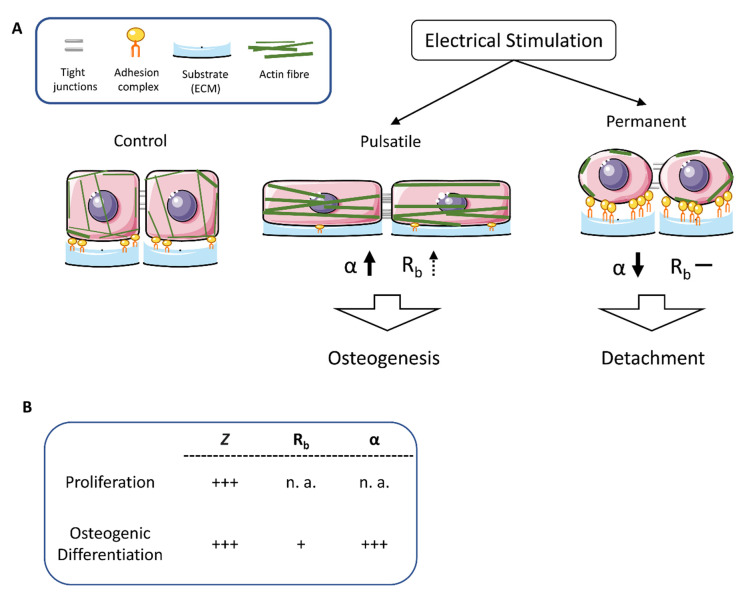
Schematic presentation of morphological alterations after pulsatile and continuous electrical stimulation. (**A**) Pulsatile stimulation resulted in decreased space between the cellular membrane and substrate and increased α values. It appears that the cells nestle against the ground, causing the actin cytoskeleton to stiffen and the number of basal adhesion proteins to decrease. In contrast, the continuous stimulation causes the cells to round up, accompanied by increased expression of the adhesion molecules in the periphery to prevent the cells from detaching. The actin fibers mainly localize to the cell periphery. (**B**) Summary of the alterations of impedance (Z), resistance (R_b_), and cleft impedance (α) in comparison of proliferating and osteogenic differentiating stem cells. + low increase, +++ significant increase, n. a. no alterations.

**Table 1 biomedicines-11-00697-t001:** List of donors from which adipose-derived stem cells were isolated. Sex, age, sampling location, and the passages used in all experiments are given.

**No.**	**Sex**	**Age**	**Sampling Location**	**Used Passages**
Z3727	female	47	upper abdomen	3–6
Z4378	female	36	abdomen	2–7
Z4169	female	56	upper abdomen	3–6
Z4054	female	55	abdomen	3–7
Z4004	male	56	abdomen	2–5
Z4058	female	49	thigh	2–6

**Table 2 biomedicines-11-00697-t002:** summarizes the differences in the impedance values of all measurements between t = 0 and t = 150 h. The values with * were significant different (*p* < 0.05) compared to controls. All chemical treatments led to significantly higher impedance in the stem cells (ASC, BMSC), and the primary osteoblasts (POB). The electrical stimulation influenced only the impedance rate of the stem cells. In brief, continuous stimulation over three days increased the impedance to approximately 100 Ω. Longer continuous stimulation times did not affect the impedance. Pulsatile stimulation, whether for 3 or 7 days, increased the impedance to approximately 70–75 Ω and 55 Ω, respectively.

	**Stimulation Conditions**		**ASC**	**BMSC**	**POB**
	Control		7 ± 6	11 ± 10	100 ± 19
	Chemical		250 ± 13 *	220 ± 11 *	150 ± 17 *
Electrical	Continuous	3 d 7 d	115 ± 32 * 4 ± 5	100 ± 19 * n.d.	96 ± 21 n.d.
	Pulsatile	3 d 7 d	70 ± 38 * 55 ± 6 *	75 ± 15* n.d.	87 ± 18 n.d.

## Data Availability

The raw data and the processed data required to reproduce these findings are available from the corresponding author upon request.

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
