# Peer review of "Combining Electrostimulation with Impedance Sensing to Promote and Track Osteogenesis within a Titanium Implant"

_biomedicines, 2023, doi:10.3390/biomedicines11030697_

Round 1

Reviewer 1 Report

This work presented by Engel et al. constructed a method that can induce osteogenesis and meanwhile permit a real-time readout of the osteogenic process. I find this work interesting, but there are quite a few technical issues that undermine this work. The language must be improved. In the methods, bullet points should be avoided. The scale bars of many images are missing or not discernable. The image quality of Fig 5B needs significant improvement. The fonts of all images need to be enlarged. Therefore, I suggest a major revision. 

Author Response

Thank you for your valuable time spent with our article, helping us to increase its scientific value! We carried out an extensive revision of our manuscript, including a language revision. See the "marked" manuscript for the exact changes.

Reviewer 1:

This work presented by Engel et al. constructed a method that can induce osteogenesis and meanwhile permit a real-time readout of the osteogenic process. I find this work interesting, but there are quite a few technical issues that undermine this work. The language must be improved. In the methods, bullet points should be avoided. The scale bars of many images are missing or not discernable. The image quality of Fig 5B needs significant improvement. The fonts of all images need to be enlarged. Therefore, I suggest a major revision.

Our response:

Two internal experts optimized the English spelling and grammar. Bullet points were excluded. Scale bars were added in Fig. 4. All other figures already had a scale bar. The resolution of Fig. 5B was increased. All fonts of the images were enlarged.

Reviewer 2 Report

The authors have performed a thorough study of what they set out to achieve. I congratulate them on their hard work. Here are some minor suggestions to further enhance the quality of their paper.

Overall suggestion: Accept with minor changes

Minor suggestions:

1.     Section 2.1 – please label your table as Table one with description. For the last donor – I think the source spelling of thigh needs correcting. Automatically, what you have labelled as Table 1 in your article will now be Table 2.

2.     Please could you provide further details on the isolation of the ASCs and the BMSCs. Were they isolated using plastic adherence or via positive selection? Currently, it is unclear.

3.     Could you please include 2-3 lines on figure 4C? Currently, I don’t see a description of the results. You could mention the fold changes between the expression of OPN between the four categories of data shown. Similar may be done for the expression of BMP-2. Were stats performed on this data set?

4.     Reflecting on the work performed – what do you think could potentially be improved in your article? More ‘n’ numbers? More cell types to investigate? What challenges do you think you will face when taking this technology forward for actual application?

5.     Also – how do you see this work being taken forward for future applications – you have mentioned this from lines 569-575. Please include these critical thoughts from the point above (point 4) along with the future perspective lines under a separate title of ‘Limitations and future directions’.

Author Response

Thank you for your valuable time spent with our article, helping us to increase its scientific value! We carried out an extensive revision of our manuscript, including a language revision. See the "marked" manuscript for the exact changes.

Reviewer 2

 Overall suggestion: Accept with minor changes

 Minor suggestions:

  1. Section 2.1 – please label your table as Table one with description. For the last donor – I think the source spelling of thigh needs correcting. Automatically, what you have labelled as Table 1 in your article will now be Table 2.

Our response:

Done.

  1. Please could you provide further details on the isolation of the ASCs and the BMSCs. Were they isolated using plastic adherence or via positive selection? Currently, it is unclear.

Our response:

Done. Please see section 2.1.

  1. Could you please include 2-3 lines on figure 4C? Currently, I don't see a description of the results. You could mention the fold changes between the expression of OPN between the four categories of data shown. Similar may be done for the expression of BMP-2. Were stats performed on this data set?

Our response:

Done, starting at line 373 as well as in the figure legend. Stats were added.

  1. Reflecting on the work performed – what do you think could potentially be improved in your article? More 'n' numbers? More cell types to investigate? What challenges will you face when taking this technology forward for actual application?
  2. Also – how do you see this work being taken forward for future applications – you have mentioned this from lines 569-575. Please include these critical thoughts from the point above (point 4) along with the future perspective lines under a separate title of 'Limitations and future directions'.

Our response:

Section 5, 'Limitations and future directions', was added. The greatest challenge of this in vitro preparatory work is the transfer to an animal model. Currently, we are facing the development of electrically stimulable electrodes into Aachener Minipigs. The electrodes and implantable stimulation system were designed in correspondence with the in silico modeling of an alloplastic, electronically active bridging system (osteosynthesis) for critical size defects of the mandible. The feasibility of using impedance testing and wireless transmission capacity could be proven in these pilot experiments that are currently analyzed for publication. The most challenging is the power supply and the range of the wireless readout. In terms of power supply, the system will be further improved by coating the titanium surface with a dielectric ceramic with piezoelectric properties such as barium titanate that could be additionally used as a transducer of the electrical current. We work with various electrical engineers and chip manufacturers to optimize signal transmission.

Round 2

Reviewer 1 Report

NA.